# The Effect of Extremely Low-Frequency Electromagnetic Fields on Inflammation and Performance-Related Indices in Trained Athletes: A Double-Blinded Crossover Study

**DOI:** 10.3390/ijms241713463

**Published:** 2023-08-30

**Authors:** Irit Markus, Evyatar Ohayon, Keren Constantini, Keren Geva-Kleinberger, Rawan Ibrahim, Angela Ruban, Yftach Gepner

**Affiliations:** 1Department of Health Promotion, School of Public Health, Faculty of Medicine, and Sylvan Adams Sports Institute, Tel-Aviv University, Tel-Aviv 69978, Israel; iritmarkus@mail.tau.ac.il (I.M.); kereng2@mail.tau.ac.il (K.G.-K.); 2Sylvan Adams Sports Institute, Tel-Aviv University, Tel-Aviv 69978, Israel; ohayon@tauex.tau.ac.il (E.O.); kconstantini@tauex.tau.ac.il (K.C.); 3Department of Biomedical Engineering, Tel-Aviv University, Tel-Aviv 69978, Israel; rawanibrahim@mail.tau.ac.il; 4Steyer School of Health Professions, Faculty of Medicine, Tel-Aviv University, Tel-Aviv 69978, Israel; 5Sagol School of Neuroscience, Tel-Aviv University, Tel-Aviv 69978, Israel

**Keywords:** recovery, extremely-low frequency electromagnetic fields, performance, inflammation, athletes

## Abstract

Previous investigations have demonstrated the therapeutic advantages of extremely low-frequency electromagnetic fields (ELF-EMFs) in mitigating inflammation and influencing biological processes. We aimed to shed light on the effects of ELF-EMF on recovery rate following high-intensity exercise. Nine male athletes (26.7 ± 6.0 years; 69.6 ± 7.7 kg, VO_2_peak 57.3 ± 6.8 mL/kg/min) completed five visits in a double-blinded crossover design, performing two consecutive testing days, following a ventilatory thresholds assessment. Following 62 min of high-intensity cycling, participants lay on an ELF-EMF mattress under active (A) and non-active (NA) conditions, immediately post protocol and during the night. Physical performance and blood markers were assessed at baseline and at 60 min (60 P) and 24 h (24 H) post-protocol. The A-condition demonstrated a notable reduction in interleukin-10 (IL-10) concentrations (mean difference = −88%, *p* = 0.032) and maximal isometric strength of the quadriceps muscles (mean difference = ~8%, *p* = 0.045) compared to the NA-condition between 60 P and 24 H. In a sensitivity analysis, the A-condition revealed that younger athletes who possessed lower fat mass experienced attenuated inflammation and biochemical responses and improved physical performance. In conclusion, ELF-EMF showed no significant overall effects on performance and inflammation after intense cycling among athletes. Post-hoc analysis revealed modest benefits of ELF-MLF, suggesting a context-dependent impact. Further research with a larger sample size and multiple sessions is needed to confirm the recovery potential of ELF-EMF.

## 1. Introduction

Professional and elite cyclists participate in diverse multistage competitions or consecutive days races (e.g., Tour de France, Giro d’Italia, Ultra Trail du Mont Blanc^®^) that require a fast and efficient recovery rate to allow optimal physical conditions for competition [1]. These populations are exposed to high physiological demands on a daily basis, which can lead to cardiovascular, immune, metabolic, endocrine, and musculoskeletal responses, ultimately resulting in the deterioration of physical capacity and performance [2]. Other important aspects of performance that are expressed during multistage races are peripheral and central fatigue, which are manifested as reduced muscle strength and power [3]. Therefore, the recovery rate of athletes during multistage competitions stands as a noteworthy concern, given limited proven protocols [4].

A non-invasive device generating extremely low-frequency (ELF) electromagnetic fields (EMFs) can be used to promote complex physiological, immune, and biomechanical effects on cells. These include facilitating vasodilatation, angiogenesis, brain plasticity, glutamate transition, and functional and mental parameters; reducing oxidative stress; and ultimately, stimulating nerve and muscle cells [4,5,6,7]. The EMFs penetrate into unshielded biological tissues where they interact with ions and proteins [8]. A previous investigation in vitro suggested that EMFs affect the expression of biological mediators, such as inflammatory cytokines, that can accelerate wound healing depending on the EMF intensity, dose, and frequency [9]. Most of the animal model studies using EMFs have shown that an ELF-EMF device inhibits the extracellular secretion of both interleukin-1 beta (IL-1β) and tumor necrosis factor-alpha (TNF-α) in culture supernatants in an intensity-dependent manner (2 Hz, 0.5–3.0 A/m) [10]. This proposed effect brings about alterations in vesicular transport and transmembrane dissociation, inducing modifications in the membrane potential.

Previous studies have demonstrated the anti-inflammatory effects of ELF-EMF exposure in human synoviocytes, chondrocytes, and osteoblasts. Namely, ELF-EMF causes a significant reduction in proinflammatory cytokines such as interleukin-6 (IL-6) and interleukin-8 (IL-8), while stimulating the release of interleukin-10 (IL-10), an anti-inflammatory cytokine (75 Hz, 1.5 mT) [6]. In addition, it has been shown that pulsed extremely low-frequency electromagnetic fields (PELF-EMFs) can modulate the passage of ions and/or the distribution of proteins through the cell membrane [11,12]. However, very few studies have evaluated the multi-system effect of ELF-EMF in humans [13,14,15]. In stroke patients, ELF-EMF has been found to reduce oxidative stress and substantially impact psychophysical abilities [16]. In addition, the immune system appears to be affected by exposure to PELF-EMFs (75 Hz, 2, 3 mT) which act to promote proteoglycan synthesis and preserve cartilage function [17]. Based on the accumulated data to date, it is reasonable to assume that exposure of trained athletes to PELF-EMF during or shortly after sports competitions will reduce the pro-inflammatory response and improve their recovery and performance-related indices.

Considering the importance to athletes of having an accelerated recovery phase following multi-day competitions and evidence of the therapeutic benefits of ELF-EMF, we aimed to examine the effect of ELF-EMF on recreationally trained cyclists’ recovery rates following acute, high-intensity exercise. Our hypothesis was that ELF-EMF would attenuate the secretion of inflammatory and biochemical markers, blunt the response of performance-related indices, and reduce subjective soreness and fatigue.

## 2. Results

Ten trained male athletes were recruited to participate in the study. However, nine (26.7 ± 6.0 years; 69.6 ± 7.7 kg, VO_2_peak 57.3 ± 6.8 mL/kg/min) completed the five study visits and one participant completed only the baseline measurements (Table 1). No significant differences were found between baseline measurements obtained during visits 2 and 4 (Table 2).

### 2.1. Blood Measurements

No significant time or condition effects were found in ΔTNF-α and CK (Figure 1A,B). A significant time effect (*p* = 0.37) was found in IL-10 changes; data for both conditions and all times points were only available for 5 participants. A significant (*p* = 0.032) elevation of ~150% was observed for the NA-condition at 60 P; however, no significant changes were found for the A-condition. Moreover, a significant (*p* = 0.025) reduction was observed for the NA-condition between 60 P and 24 H (Figure 1C).

### 2.2. Performance-Indices Assessment

No significant effects were found for ΔTW, ΔPPT, or ΔPP DEC between or within groups over time (Figure 2A–C). However, a significant time effect (*p* = 0.018) was observed in MVC changes. A decrease of ~10% at 60 P was observed for both groups. At 24 H, the A-condition showed a return to baseline but the NA-condition still remained ~5% lower than BL (Figure 2D). When the data were stratified according to the study conditions, a significant increase in MVC was found for the A-condition between 60 P and 24 H (*p* = 0.045), but not for the NA-condition (*p* = 0.158).

### 2.3. Subjective Measurements

In both the VAS and ROF, no significant condition or time effects were observed. The ROF level slightly (32%) but not significantly (*p* = 0.44) increased at 60 P for both groups. Although not statistically significant, the ROF level for the A-condition remained at a similar level at 24 H (27% compared to BL), while that for the NA-condition continuously increased (46% compared to BL), with *p* of interaction = 0.93. A similar pattern was observed in the VAS, but the VAS for the NA-condition remained higher at 24 H (47% compared to BL), while that for the A-condition was attenuated (9%).

### 2.4. Sensitivity Analysis

In a sensitivity analysis of a hierarchical cluster, we found significant (*p* < 0.05) differences within the cohort in age (cluster 1, *n* = 4: 21.7 ± 3.3 years; cluster 2, *n* = 5: 32.4 ± 1.3 years) and fat mass (cluster 1: 6.7 ± 2.5 kg; cluster 2: 11.6 ± 1.37 kg). Then, we added another layer into the analysis and found a stronger response to the mattress within cluster 1. Cluster 1 also showed higher but not significant values of VO_2_peak (60.2 ± 4.4 mL/kg/min) compared to cluster 2 (55.0 ± 7.9 mL/kg/min) (Appendix A).

### 2.5. Blood Measurements Related to the Cluster Analysis

No significant effects were found in TNF-α for the A- and NA-conditions (Figure 3A). A significant (*p* = 0.039) interaction was found between condition × cluster in CK. Specifically, the CK concentration for the NA-condition was significantly higher than that for the A-condition within cluster 1 at 24 H (*p* = 0.001; Figure 3B). Significant time (*p* = 0.027) and condition (*p* = 0.038) effects were observed in IL-10 changes. Specifically, the IL-10 concentration showed a significant elevation from BL for the NA-condition compared to the A-condition within cluster 1 at 60 P (*p* = 0.031; Figure 3C).

### 2.6. Performance Assessment

No significant changes were found in PP, DEC, or MVC (Figure 4B,D). A significant interaction between condition × cluster (*p* = 0.13) and a significant time effect (*p* = 0.015) were found in TW. A significant (*p* = 0.044) elevation was found for the A-condition between 60 P and 24 H within cluster 2; however no differences were found within cluster 2. Based on the stratification to clusters, significant (*p* = 0.008) differences were found between clusters 1 and 2 for the A-condition at 60 P (Figure 4A). In addition, a significant interaction between condition × cluster was found in PPT (*p* = 0.028). Specifically, based on the stratification to clusters, a significant (*p* = 0.024) difference was found between clusters 1 and 2 for the A-condition at 60 P (Figure 4C).

### 2.7. Subjective Measurements

No significant differences between conditions and clusters were found in ROF and VAS (Appendix A). The ROF level slightly decreased numerically within cluster 1 for the A-condition (−7%), while for the NA-condition, the ROF level was numerically higher compared to baseline at 24 H (70%), *p* = 0.89. For both conditions, cluster 2 showed increased levels of ROF. For the A-condition within cluster 1, the VAS remained stable (−0.7%) at 24 H, while within cluster 2, it increased non-significantly (25%). For the NA-condition within cluster 1, the VAS increased (66%) at 24 H, while within cluster 2, it increased to a lesser extent (22%), but neither of these increases were significant.

## 3. Discussion

To our knowledge, this is the first study to investigate the effect of ELF-EMF on inflammation and performance-related indices in trained athletes. Specifically, we examined the effects of ELF-EMF stimulation on the rate of recovery following a single bout of high-intensity exercise by looking at physiological indices that included physical performance, biochemical and inflammatory markers, and the subjective perception of pain and rate of fatigue in trained cyclists. We found improvements in the rate of recovery in a few parameters following ELF-EMF stimulus, particularly among younger athletes with low fat mass. Therefore, we suggest that ELF-EMF may potentially blunt the reduction in muscle strength during high-intensity daily exercise by expediting the recovery process.

IL-10 concentrations were significantly blunted by ELF-EMF stimulation between 1 and 24 h following the high-intensity protocol (60 P and 24 H) for the A-condition. Recovery, which allows physiological systems to return to homeostasis between training sessions, is an essential state for optimal adaptation to exercise [18]. The acute-phase response and cytokine release during this recovery phase are normal and beneficial for the body’s adaptation to exercise. However, when inflammation becomes higher or persistent as a result of continuous physical-exercise effort, it can affect an athlete’s performance-related indices [19]. The rate of recovery is influenced by the immune system (e.g., IL-10, IL-6, and TNF-α) and biochemical responses (e.g., CK), as they promote the recruitment and activation of immune cells to repair damaged tissues [20]. The ultimate goal of sufficient recovery is to return to baseline physical performance or even achieve improved performance following physical exercise [21]. Previous studies have demonstrated that ELF-EMF stimulation can reduce and inhibit the secretion cascade of both pro- and anti-inflammatory cytokines [6,7,22,23,24]. While our results did not show a significant condition effect for TNF-α and CK following ELF-EMF stimulation, the observed IL-10 attenuation for the A-condition may indicate an accelerated rate of recovery in the short-term, which may play a relevant role in multi-stage racing competitions.

The change in maximal voluntary contraction (MVC), a highly validated isometric assessment, returned to baseline only for the A-condition following the high-intensity protocol. It has been previously suggested that long-duration cycling is associated with neural input and muscular fatigue, which is assessed by a reduction in MVC torque [25,26]. In our study, despite having participants cycle for only 1 h, we showed that MVC returned to baseline for the A-condition in the 24 h following the exercise protocol, which may reflect central recovery. The isometric maximal force of one muscle group has been shown to be a reliable assessment of whole-body strength, which may predict an athlete’s ability to maintain high-intensity exercise for short periods of time [27]. Continuous and repetitive exercise during multi-stage racing can result in cumulative fatigue. This fatigue can be both peripheral, affecting the muscles, and central, affecting the central nervous system. Accumulated muscle damage, depletion of energy stores, and alterations in neuromuscular function contribute to fatigue and reduced performance [28]. Thus, prolonged exposure to extremely high exercise loads and an imbalanced ratio with the recovery phase, as experienced by competitive athletes, can lead to fatigue and maladaptation [29]. In competitions that span several consecutive days, it is crucial to maintain sufficient physiological demand and muscle capacity, which are reflected in high physical performance [30]. Therefore, our study finding suggests that ELF-EMF stimulation may be effective in muscle recovery and boosting performance. We did not find a significant effect on other performance indices, i.e., TW, PPT, and PP DEC. However, this is the first study to present use of ELF-EMF on recovery and performance benefits for trained athletes. Future larger studies are warranted to investigate changes in whole-body athletic performance following ELF-EMF treatment.

We performed a sensitivity analysis using hierarchical clustering techniques to mitigate the substantial variability in the participating athletes’ ages and body composition, as is often observed within athletic cohorts. It has been previously suggested that fat mass is negatively associated with muscle strength and aerobic capacity [31]. In addition, aging is associated with declining physical endurance due to various physiological and behavioral changes [32]. Based on those two key factors, we stratified the study participants into two distinct groups based on fat mass and age. We found that the application of ELF-EMF stimulation predominantly conferred advantages in younger athletes with lower fat mass. Specifically, differences were found in blood markers (e.g., IL-10, CK) and performance measurements (e.g., TW and PPT). This suggests that with younger age, ELF-EMF could possibly be more efficient physiologically. Indeed, this is in line with previous investigations that found that the effect of pulsed electromagnetic fields is age-dependent [33]. These observations, therefore, emphasize that ELF-EMF does not affect all individuals equally, and some athletes might benefit more from this treatment.

### Limitations of the Study

The study has several limitations. First, there was a relatively brief 24 h follow-up period subsequent to the high-intensity exercise protocol. This was because we wanted to emulate practical scenarios of ELF-EMF mattress usage (active intervention), such as after a race day or multi-day competition and assess the device’s effectiveness in terms of immediate post-race short exposure. However, this restricted duration might not adequately capture the entirety of the potential physiological and biochemical shifts that could manifest over a more extended timeframe. Therefore, a lengthier follow-up span of several days may provide more information. This links to the second study limitation. Namely, the study design included a constraint arising from the athletes’ competitive commitments, thereby limiting magnetic resonance exposure following the high-intensity protocol. Although this design constraint was driven by the athletes’ competitive engagements, it might have limited the effect size. In future investigations, it might prove beneficial to extend both the magnetic intervention period and subsequent follow-up window. Lastly, while all data were available and included in the final analysis for most study outcomes, the sample size was smaller for IL-10 due to the performance of the ELISA kit.

## 4. Material and Methods

### 4.1. Study Population

Ten healthy competitive male cyclists and triathletes were recruited to participate in the study. All participants were recruited based on their personal ability to perform 1 h of cycling at an average power of at least 250 W. The exclusion criteria included smokers and those taking prescribed medications or having a self-reported history of chronic pulmonary, cardiac, metabolic, or orthopedic conditions.

### 4.2. Study Design

The clinical study protocol was approved by the Ethical Committee of Tel Aviv University (#0003017-4). The participants visited the laboratory on five occasions and used their own personal bike for all of the exercise tests (Figure 5). On visit 1, the participants completed informed consent, health, and physical activity level questionnaires and anthropometric measurements (weight, height, and fat mass) were performed. They also completed a graded cycling exercise test to volitional exhaustion, familiarization trials for the maximal voluntary contraction (MVC) of the quadriceps muscles, and the repeated sprint test (RST). Visits 2 and 3 and visits 4 and 5 were performed on consecutive days with an interval of two weeks between each pair of consecutive testing days. On both pairs of days, testing commenced at the same time of day in the same order: weighing the participants, blood draws, MVC, RST, and questionnaires about their subjective perception of soreness and fatigue. These procedures were performed before the protocol at baseline (BL) and at 60 min (60 P) and 24 h (24 H) post protocol. These specific time points were chosen in order to evaluate the recovery rate over time before and following magnetic intervention up to 24 h. Athletes were asked to refrain from caffeine, alcohol, and intense physical activity 12 h prior to testing. The participants were also asked to maintain the same diet before and during visits 2 and 3 and visits 4 and 5.

During visits 2 and 4, the athletes lay supine on an ELF-EMF mattress at two separate time points: immediately post protocol for 45 min (SPR1) and during their night sleep for 6 h (SPR2). In a randomized and double-blinded manner, two different conditions were applied for each participant: active (A-condition) or non-active (NA-condition). The active condition was when the ELF-EMF mattress was emitting electromagnetic fields (Figure 5). The rationale behind the placement of the chosen timeframes was to facilitate an examination of the protocol’s impact both with and without the treatment subsequent to the high-intensity exercise regimen. The primary objective was to assess the effectiveness of the magnetic resonance intervention in a scenario that could be practical in the real world (the active condition), such as after a race day or multi-day competition.

Anthropometric assessment. Body mass, fat mass, and fat-free mass were measured using bioelectrical impendence analysis (BIA; SECA© Medical Body Composition Analyzer, mBCA 515, Hamburg, Germany). Height was measured using a stadiometer (SECA 274). Prior to this test, the participants were asked to refrain from eating and drinking for at least 2 h.

Maximal graded test. This test aimed to evaluate the maximal O_2_ uptake (VO_2_peak), which is the gold standard for maximal aerobic capacity and cardiorespiratory fitness. A Cyclus2 ergometer (Leipzig, Germany) was used for the graded test and all proceeding cycling tests. A 5 min warm-up was performed at a freely selected power output. The test began at 100–150 Watts and the power output was then increased by 25 Watts every 4 min until the second ventilatory threshold (VT2) was identified. Thereafter, the power output was increased by 25 Watts every minute until volitional exhaustion. The test was terminated when participants (voluntarily) expressed an inability to continue exercising despite strong verbal encouragement. Ventilatory and metabolic measurements were collected during the graded protocol using breath-by-breath analysis (Quark Cardiopulmonary Exercise Testing, Cosmed, Rome, Italy) while participants breathed through an oro-nasal facemask (7450 Series, Hans Rudolph, Kansas City, MO, USA). Heart rate (HR) was continuously monitored using a chest strap (Polar Electro Oy, Kampele, Finland). Peak oxygen consumption (VO_2_peak) was determined as the highest 30 s average O_2_ uptake (VO_2_) achieved during exercise while meeting two of the following three criteria: (1) a HR ≥ 90% age-predicted maximum, (2) an RER ≥ 1.10, and (3) a plateau in VO_2_ ≤ 150 mL with increased workload [34]. The maximal HR was determined as the highest HR recorded during the test. Both the first and second ventilatory thresholds (VT1 and VT2, respectively) for each participant were determined using combined methods [35].

Exercise (cycling) protocol. On visits 2 and 4, participants completed a 62 min high-intensity interval protocol that was designed to mimic a hard training session. The work-recovery ratio was 1:1, where work intensity varied relative to VT2 (see below), and recovery was set to 90%VT1. The protocol started with a 5 min warm-up at a freely chosen intensity and 5 min at VT1. All loads during the protocol were determined relative to VT2, with an ascending pyramid structure in which participants started at 120%VT2 for 30 s, 110%VT2 for 1 min, 105%VT2 for 2 min, 100%VT2 for 3 min, 95%VT2 for 4 min, and 90%VT2 for 5 min. Then, the pyramid was reversed until reaching 120%VT2 for 30 s.

Repeated sprints test (RST). This test aimed to assess the ability of the athletes to generate maximal power once and repetitively [36]. Participants were instructed to perform an all-out effort in each of the sprints by pedaling as hard and as fast as they could, while not conserving energy within or between efforts. They were instructed to stay seated and hold the handlebar in the same position during the test. Prior to each test, participants began the session with a 5 min warm-up and two sprints of 3 s each, with 1 min rest in between. Then, each participant performed 5 all-out sprints of 6 s with a resistance of 10% of their body mass. The resting time between sprinting efforts was 24 s [36]. The peak power total (PPT; i.e., the sum of peak power outputs from all the sprints), total work (TW) performed over the 5 sprints, and peak power decrement (PP DEC) were measured. PP DEC was determined as a percentage. HR was monitored continuously during the tests.

Circulating blood markers. Blood samples were obtained from the participants in order to evaluate the inflammatory and biochemical changes that occurred in response to the cycling protocol with and without ELF-EMF stimulation. The samples were obtained at three time points throughout each set of 2 consecutive testing days of the study (BL, 60 P, and 24 H). The blood samples were collected with single-use disposable needles. A total of 10 mL of blood was collected at each time point in a 10 mL Vacutainer^®^ serum collection tube in which the blood was allowed to clot. The serum was kept at room temperature for 1 h and then centrifuged at 1300× *g* for 10 min. Serum samples were placed into separate 1.8-mL microcentrifuge tubes frozen at −80 °C. Serum concentrations of IL-10 and TNF-α were analyzed using a high-sensitivity cytokine multiplex assay (Luminex, Cat no. FCSTM09-02; R&D Systems, Inc., Minneapolis, MN, USA) on a MAGPIX instrument (Luminex, Austin, TX, USA), according to the manufacturer’s instructions. Creatine kinase (CK) concentrations were analyzed using Roche clinical chemistry and immunochemistry analyzer (Cobas c 111, Indianapolis, IN, USA).

Maximal voluntary contraction test. This test aimed to measure the maximal isometric force generated by the quadriceps muscles. Participants were familiarized with this test on visit 1 and then were asked to complete the test at all visits. Participants performed three maximal voluntary isometric knee extension contractions (MVC) with 1 min rest between each of the three trials. Lying on a treatment bed in the supine position, a padded strap was secured around their right ankle with the other side of the strap secured to a metal rod in the bed, with the knee fixed at a 90° angle. Participants were instructed to extend their right knee as forcefully as possible for 5 s. Force data were collected by the strain gauge (Chronojump^®^, Barcelona, Spain) and recorded using Chronojump^®^ software (version 1.9.0) at a sampling frequency of 80 Hz. The highest value obtained during the three trials was used.

#### Questionnaires

Perception of pain in the quadriceps muscle: Prior to all performance tests, participants were asked to quantify their degree of soreness in the quadriceps muscle of their right thigh on a 15 cm visual analog scale (VAS) by making a mark on a horizontal line that ranged from “not sore at all” to “maximal soreness” anchored at each end. They were also asked to rate their quadriceps pain on the VAS immediately after performing the MVC test.

Rate of fatigue (ROF) scale: Prior to all performance tests and following the VAS questionnaire, participants were asked to rate their level of fatigue, which was measured on an 11-point numerical scale with empirically derived accompanying descriptors and diagrammatic components of the ROF [37].

### 4.3. Pulse Extremely Low Frequency—Electromagnetic Field (PELF-EMF)

A SEQEX^®^ device (S.I.S.T.E.M.I. Srl., Trento, Italy; certified CSQ ISO-13485) was used to produce complex pulsed electromagnetic fields. It used an analog mechanism with a frequency range of 1 to 80 Hz and intensities from 1 to 20 µT. The electromagnetic field produced by the device control unit (on which the electromagnetic field parameters were set) was emitted from a mat containing a Helmholtz coil that generated the PELF-EMF.

SPR1: Following the exercise protocol, participants were asked to lay supine for 45 min on a portable electro-magnetic mattress (Table 3a). At visits 2 and 4, they received an active (A) card or non-active (NA) sham card by random assignment. Another electrode device, Octopus (SEQEX^®^, Trento, Italy), with eight cords, was placed on the bellies of the four muscles of interest on each leg; the quadriceps, hamstring, gastrocnemius, and tibialis anterior. This was performed in order to increase the electromagnetic field pulses. Participants were instructed to remove and distance all personal electronic devices and jewelry during the procedure.

SPR2: Participants were instructed to have a regular night sleep for at least 6 h during the nights between visits 2 and 3 and visits 3 and 4 and lay on the portable electro-magnetic mattress at their homes (Table 3b). Participants were given a similar activating card (A- or NA-condition) as they used during SPR1 of the same visit.

### 4.4. Study Rationale and Operating Hypotheses

According to Liboff and Zhadin, the proposed explanation of the biological mechanism of ELF-EMF is that coupling of a variable EMF (in the range applied in this study) with a geomagnetic field (GMF) creates the conditions for an ion cyclotron resonance-like (ICR-like) effect [18,38]. Indeed, according to in vivo and in vitro studies, ELF-EMF frequencies have been demonstrated to induce various biochemical and physiological responses [10,19,20,21,22,23,24,25,26,27,28,29]:2 Hz affects inflammation by downregulating TNF-α and IL-1β;4 Hz reduces oxidative stress;12 Hz improves local microcirculation;15 Hz increases alkaline phosphatase activity (ALP) and chondrogenesis;30 Hz affects inflammation by downregulating IL-10;50 Hz impacts inflammation by reducing chemokine production;75 Hz upregulates A2A and A3 adenosine receptors and induces anti-inflammatory and antioxidant effects.

ELF-EMF frequencies were chosen for the study by considering the above findings from previous studies, and accordingly the hypotheses for SPRI and SPR2 were:

SPR1:(a)Concentrations of TNF-α and IL-1β, which are highly produced in the first hours after a workout, will be reduced using a frequency of 2 Hz.(b)Local oxidative stress, strictly linked with inflammation, will be reduced using a frequency of 4 Hz.(c)ALP activity will be stimulated using a frequency of 15 Hz.

SPR2:(a)Local inflammation will be modulated with frequencies of 75 and 50 Hz.(b)Local microcirculation will be stimulated using a frequency of 12 Hz.(c)Levels of cytokines TNF-α and IL-1β (using 2 Hz) and IL-10 (using 30 Hz) will be reduced.

### 4.5. Statistical Analysis

The participants’ characteristics are presented as mean ± standard deviation. Repeated measures analysis of variance (ANOVA) was used to compare performance-related indices and subjective assessments between condition and time. Changes from baseline (∆%) of blood markers (e.g., CK, IL-10, and TNF-α), physical performance-indices (MVC, TW, PPT, and PP DEC), and subjective assessment (ROF and VAS) between A- and NA-conditions were analyzed using two-factor (time × condition) ANOVA. In the event of a significant F-ratio, Bonferroni post-hoc analysis was used for pairwise comparisons. In a sensitivity analysis, the participants were stratified using hierarchical clustering based on age and fat mass into two distinct groups to better understand the effect of ELF-EMF stimulation on performance and strength during the recovery phase in a sub-group of athletes. Baseline characteristics between clusters were compared using the Mann–Whitney test. Relative changes between study time points, conditions, and clusters were analyzed using three-factor (time × condition × cluster) multivariate analysis of variance (MANOVA). Significant changes were considered as alpha of *p* ≤ 0.05. All data were analyzed using IBM SPSS Statistics v25 software (SPSS, Inc., Chicago, IL, USA).

## 5. Conclusions

In summary, our study did not find statistically significant overall effects of ELF-EMF on performance and blood markers of inflammation following single high-intensity cycling among highly trained athletes. However, post-hoc analyses revealed modest yet noteworthy benefits, indicating that the impact of ELF-EMF might be context specific. These findings suggest potential positive effects in specific scenarios. Future research endeavors should aim to enhance the robustness of the findings regarding the advantageous effects of ELF-EMF as a recovery strategy by incorporating a more extensive sample size and conducting multiple exercise sessions.

## Figures and Tables

**Figure 1 ijms-24-13463-f001:**
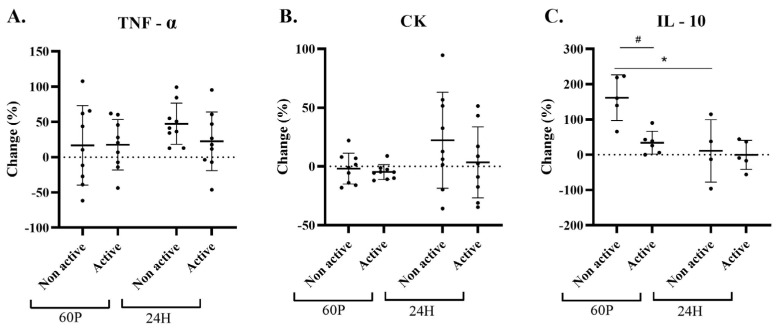
Changes in blood markers of muscle damage and inflammation: tumor necrosis factor-α [TNF-α; panel (A)], creatine kinase [CK; panel (B)], and interleukin-10 [IL-10, *n* = 5; panel (C)] relative to baseline (BL). Active label represents A-condition and Non active label represents NA-condition. Measurements were taken at BL and at 60 min (60 P) and 24 h (24 H) following a 60 min cycling pyramid protocol. Two-factor (time × condition) repeated measures analysis of variance (ANOVA) with Bonferroni correction was performed. * Significant (*p* < 0.05) time effect stratified by condition. # significant condition effect for the change at 60 P in the post-hoc analysis. All data are reported as mean ± SD.

**Figure 2 ijms-24-13463-f002:**
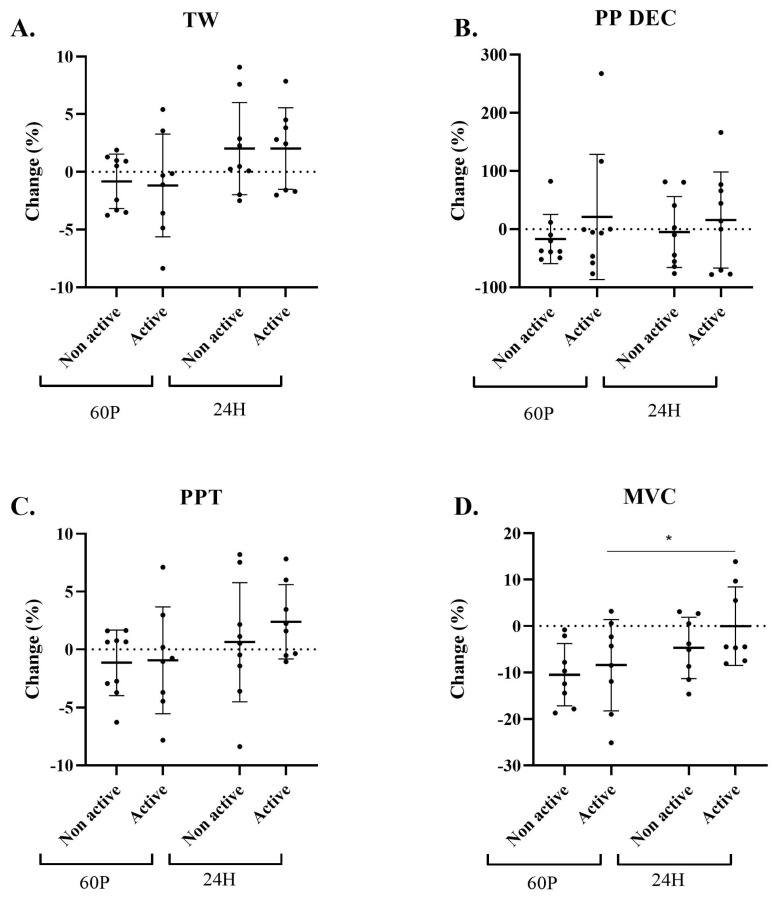
Changes in total work [TW, panel (A)], peak power decrement [PP DEC, panel (B)], peak power total [PPT, panel (C)], and maximal voluntary contraction [MVC, panel (D)] relative to baseline (BL). Active label represents A-condition and Non active label represents NA-condition. Cadence and stride length were recorded while running at a constant pace of 9 km/h. Measurements were taken at BL and at 60 min (60 P) and 24 h (24 H) following a 60 min cycling pyramid protocol. Two-factor (time × condition) repeated measures analysis of variance (ANOVA) with Bonferroni correction was performed. * Significant (*p* < 0.05) time effect stratified by condition. All data are reported as mean ± SD.

**Figure 3 ijms-24-13463-f003:**
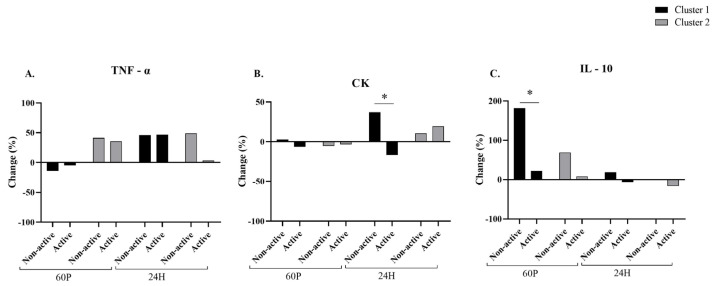
Changes in blood markers of muscle damage and inflammation by clusters: tumor necrosis factor-α [TNF-α; panel (A)], creatine kinase [CK; panel (B)], interleukin-10 [IL-10 (*n* = 5); panel (C)] relative to baseline (BL). Active label represents the A-condition and Non active label represents the NA-condition. Measurements were taken at BL and at 60 min (60 P) and 24 h (24 H) following a 60 min cycling pyramid protocol. Black bars represent cluster 1, gray bars represent cluster 2. Two-factor (time × condition) repeated measures analysis of variance (ANOVA) with Bonferroni correction was performed. * Significant (*p* < 0.05) condition effect, stratified by cluster. All data are reported as mean ± SD.

**Figure 4 ijms-24-13463-f004:**
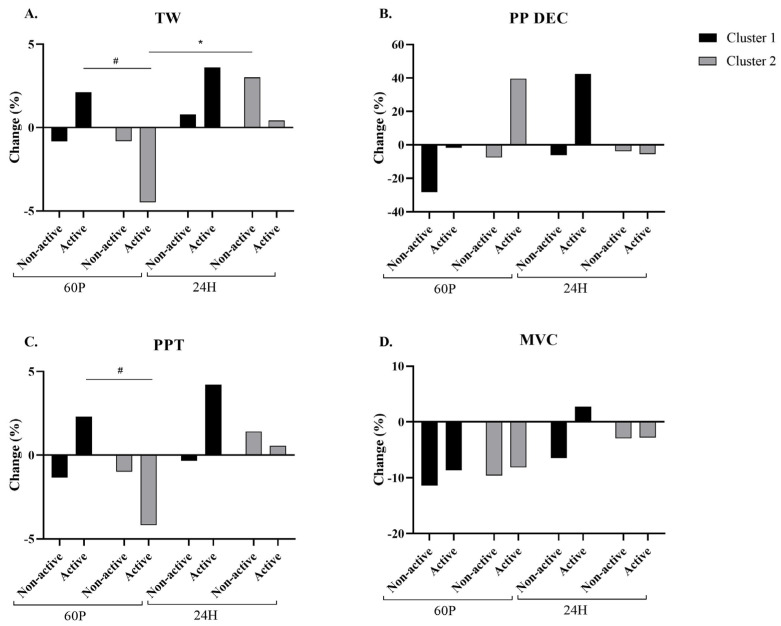
Changes in total work [TW, panel (A)], peak power decrement [PP DEC, panel (B)], peak power total [PPT, panel (C)], and maximal voluntary contraction [MVC, panel (D)] relative to baseline (BL). Active label represents the A-condition and Non active label represents the NA-condition. Cadence and stride length were recorded while running at a constant pace of 9 km/h. Measurements were taken at BL and at 60 min (60 P) and 24 h (24 H) following a 60 min cycling pyramid protocol. Black bars represent cluster 1, gray bars represent cluster 2. Two-factor (time × condition) repeated measures analysis of variance (ANOVA) with Bonferroni correction was performed. * Significant (*p* < 0.05) time effect stratified by condition, # Significant (*p* < 0.05) cluster × condition stratified by time. All data are reported as mean ± SD.

**Figure 5 ijms-24-13463-f005:**
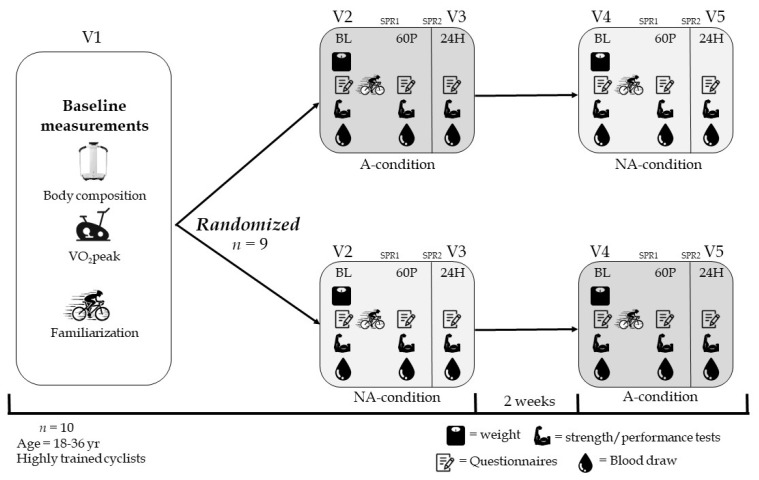
The experimental protocol. V, visit; BL, baseline; 60 P, 60 min post-protocol; 24 h, 24 H post protocol; VO_2_peak, O_2_ peak consumption; Body composition, including weight, height, body mass, fat mass, and fat-free mass; Familiarization, including maximal voluntary contraction and repeated sprint test; A-condition, active condition; NA-condition, non-active condition. SPR1 was located at V2 and V4 immediately following the exercise cycling protocol and SPR2 was located at night between V2 and V3 and between V4 and V5.

**Table 1 ijms-24-13463-t001:** Baseline characteristics of the study population.

Characteristics	Findings (*n* = 9)
Age (y)	26.7 ± 6.0
Height (m)	1.77 ± 0.06
Weight (kg)	69.6 ± 7.7
Fat mass (%)	13.4 ± 3.7
Fat-free mass (kg)	60.2 ± 5.7
HRmax (bpm)	184 ± 11
VO_2_peak (mL/min)	3965 ± 400
VO_2_peak (mL/kg/min)	57.3 ± 6.8
Maximal power (W)	400 ± 35
Power at VT1 (W)	216 ± 39
Power at VT2 (W)	281 ± 40

VT1, first ventilatory threshold; VT2, second ventilatory threshold. All data are reported as mean ± SD.

**Table 2 ijms-24-13463-t002:** Baseline performance and subjective assessment of each condition.

	Active (*n* = 9)			Non-Active (*n* = 9)			*p* Value
	BL	60 P	24 H	BL	60 P	24 H	
MVC (Newton)	467.5 ± 71.9	425.1 ± 58.3	464.2 ± 53.6	478.9 ± 67.9	425.1 ± 36.9	456.6 ± 77.4	0.951
TW (Joule)	22,034 ± 3103	21,706 ± 2631	22,427 ± 2845	21,705 ± 3404	21,508 ± 3236	22,164 ± 3711	0.998
PPT (Watts)	3904 ± 621	3857 ± 561	3989 ± 578	3944 ± 783	3889 ± 699	3955 ± 689	0.984
PP DEC (%)	3.69 ± 3.33	3.45 ± 2.91	3.22 ± 2.38	4.59 ± 2.04	3.58 ± 1.69	3.90 ± 2.46	0.898
ROF (0–11 scale)	2 ± 1	2 ± 1	2 ± 2	2 ± 1	2 ± 1	3 ± 2	0.925
VAS (0–15 scale)	4 ± 3	6 ± 4	4 ± 3	2 ± 2	4 ± 3	4 ± 2	0.838

MVC, maximal voluntary contraction, TW, total work; PPT, peak power total; PP DEC, peak power decrement; ROF, rate of fatigue; VAS, visual analog scale. Independent sample *t*-tests were performed. All data are reported as mean ± SD.

**Table 3 ijms-24-13463-t003:** (**a**) SPR1: The 45 min PELF-EMF treatment protocol. (**b**) SPR2: The 6 h PELF-EMF treatment protocol.

Step	Waveform	Frequency (Hz)	Intensity (%)	T-on (s)	T-off (s)	Duration (min)
(**a**)						
1	14	2	100	4	1	5
2	25	30	100	3	2	7
3	18	4	100	3	1	5
4	7	50	100	3	1	3
5	21	2	100	2	1	7
6	25	15	100	3	1	3
7	26	75	100	3	2	7
8	26	50	100	5	2	5
9	29	20	100	5	1	3
(**b**)						
1	20	75	100	3	2	40
2	14	50	100	5	2	40
3	5	12	100	2	1	40
4	25	20	100	5	1	40
5	26	2	100	4	1	40
6	27	4	100	3	1	40
7	29	30	100	3	2	40
8	20	50	100	5	2	40
9	19	2	100	4	1	40

## Data Availability

The data is available on request from the corresponding author.

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
