# Peer review of "The Effect of Extremely Low-Frequency Electromagnetic Fields on Inflammation and Performance-Related Indices in Trained Athletes: A Double-Blinded Crossover Study"

_ijms, 2023, doi:10.3390/ijms241713463_

Round 1
Reviewer 1 Report
Small but well designed study to document the effectiveness of an EMF generating mat for athletes.
Work appears to be carefully done but I have significant concerns about the documentation that can be addressed by major revisions.
Mail problem is the extensive post-hoc character of the work, which casts doubt on the validity of the conclusions.
Following the recommendations of Michel at al (Drug Metab Dispos 48:64—74, January 2020; https://dmd.aspetjournals.org/content/48/1/64) would the authors please:
1. state which parts (if any) of the study test a hypothesis according to a prewritten protocol and which parts are more exploratory.
2. Include quantitative indications of effect sizes and confidence intervals in the abstract. p values are not effect sizes!!
Also need to correct the statistics for multiple comparisons (Bonferroni etc.)
On looking at the data, I see lots of comparisons, only a few of which are statistically significant, and it looks to me like the study as a whole was negative if multiple comparison artifacts are properly addressed. Is that correct?
The data plots show fewer than 10 data points -- does this mean that some data were excluded? If so indicate the rationale for dropping data.
The conclusion (sec 5) is weak. Study basically finds that the effects, if there are any, in most cases are too small to be resolved from the noise. If the effect sizes are too small to be physiologically meaningful (as I gather) please state so directly. If the authors see the possibility of physiologically significant effects it is OK to discuss that but justify your reasoning. Please don't just write that EMF "could possibly" benefit "some performance outcomes".
minor:
line 239 says "nine trained athletes" but the table indicates n=10. Error?
please clarify in the figures, does "active" mean that the mat was generating magnetic fields? (I think so but am not sure). Add that to a figure caption.
Should not results with an inactive mat at any particular time be the proper control? If that is the case, then the study is almost entirely negative.
Please don't use the term "significant" without a proper modifier to remove ambiguity. I think you mean "statistically significant" but readers are likely to understand that as "physiologically significant" - not proven here.
in the introduction, the list of other studies reporting "effects" of ELF-EMF, without stating exposure strength or magnitude, is not informative. Better to replace this with a table listing the studies, endpoint, exposure characteristics, outcome.
Author Response
We would like to thank you for handling the resubmission of our manuscript, entitled "The effect of extremely low-frequency electromagnetic fields on inflammation and performance-related indices in trained athletes: a double-blinded crossover study". We are grateful to the reviewers for their enlightening comments and important suggestions. In the revised version of the manuscript, we have addressed the specific points raised in their reviews, as described in more detail below in bold text and within the manuscript in red. We are confident that the alterations made in response to the reviewers' input will serve to elevate the overall quality and impact of our study.
Reviewer #1
Small but well designed study to document the effectiveness of an EMF generating mat for athletes. Work appears to be carefully done but I have significant concerns about the documentation that can be addressed by major revisions.
Response: We were very encouraged that the manuscript was found to be “well designed and carefully done”, and we are highly appreciative of the reviewer’s comments and their clear instructions on how to revise and improve the manuscript.
Mail problem is the extensive post-hoc character of the work, which casts doubt on the validity of the conclusions.
Response: We thank the reviewer for the comment. In accordance with your suggestion, we modified the post hoc analysis to Bonferroni for all the relevant statistical analysis and revised the conclusion paragraph within the abstract and the main body of the manuscript to more accurately portray the study's findings. The specific alterations we implemented are as follows:
Abstract (page 1, lines 27-31):
“In conclusion, ELF-EMF showed no significant overall effects on performance and inflammation after intense cycling among athletes. Post-hoc analysis revealed modest benefits of ELF-MLF, suggesting a context-dependent impact. Further research with larger samples and multiple sessions is needed to confirm ELF-EMF's recovery potential.”
Conclusion (page 13, lines 466-473):
“In summary, our study did not find statistically significant overall effects of ELF-EMF on performance and blood inflammation following single high-intensity cycling among highly trained athletes. However, post-hoc analyses revealed modest, yet noteworthy benefits, indicating that the impact of ELF-EMF might be context-specific. These findings suggest potential positive effects in specific scenarios. Future research endeavors should aim to enhance the robustness of the findings regarding the advantageous effects of ELF-EMF as a recovery strategy by incorporating a more extensive sample size and conducting multiple exercise sessions.”
Following the recommendations of Michel at al (Drug Metab Dispos 48:64—74, January 2020; https://dmd.aspetjournals.org/content/48/1/64) would the authors please:
state which parts (if any) of the study test a hypothesis according to a prewritten protocol and which parts are more exploratory.
Response: We agree with the reviewer’s suggestion. The following hypothesis have been added to the manuscript:
“Our hypothesis was that ELF-EMF would attenuate the secretion of inflammatory and biochemical markers, blunt the response of performance-related indices, and reduce subjective soreness and fatigue.” (page 2, lines 78-80). In addition, the introductory part includes a detailed description of the stimulation frequencies reported in the literature and their effects on various biochemical parameters included in our research protocol.
In addition, the rationale and operating hypotheses of the electromagnetic mattress was based on a vast literature review that cited in the methods materials which including more than twelve reference. (page 6 lines 230-235):
“According to Liboff and Zhadin the proposed explanation of the biological mechanism of ELF-EMF is that the coupling of a variable EMF (in the range applied in this study) with the geomagnetic field (GMF) creates the condition for the ion cyclotron resonance-like (ICR-like) effect [22,23]. Indeed, according to in vivo and in vitro studies, ELF-EMF frequencies have been demonstrated to induce various biochemical and physiological responses [10,24–34]”.
Include quantitative indications of effect sizes and confidence intervals in the abstract. p values are not effect sizes!!
Response: Thank you, we have added quantitative information to the abstract (page 1, lines 24-25) as follows:
“The A-condition demonstrated a notable reduction in IL-10 (mean difference = -88%, p=0.032) and maximal isometric strength of the quadricep muscles (mean difference = ~8%, p=0.045) compared to the NA-condition between 60P and 24H.”
Also need to correct the statistics for multiple comparisons (Bonferroni etc.)
Response: We appreciate your comment, and accordingly, we have added to the results section a post hoc analysis with Bonferroni correction. The results section, including text and figures is now completely modified with the new statistical analysis, which contains Bonferroni corrections (pages 8-9, lines 288-374).
On looking at the data, I see lots of comparisons, only a few of which are statistically significant, and it looks to me like the study as a whole was negative if multiple comparison artifacts are properly addressed. Is that correct?
Response: We appreciate the comment and understand the reviewer’s concern. Our per-protocol analysis was performed to determine main and time effects in the dependent outcomes. As mentioned, the revised analysis, using Bonferroni corrections, control for multiple comparisons or corrections in the statistical analyses.
The data plots show fewer than 10 data points -- does this mean that some data were excluded? If so indicate the rationale for dropping data.
Response: Thank you for this valuable comment. It is important to note that no data points were deliberately excluded from the analysis. However, it should be highlighted that certain blood markers, notably IL-10, exhibited incomplete data across all time points due to limitations in the performance of the ELISA kit. This limitation impeded our ability to calculate temporal changes accurately. To address this concern, we have incorporated this pertinent information into the captions of the figures, and the result section, as needed. We also wrote in section 3.1: “A significant time effect (p = 0.37) was found for IL-10 changes; data for both conditions and all times points were only available for 5 participants.” In addition, we have added the available sample size of IL-10 in the figure’s legends (Fig 2 and Fig 4).
The conclusion (sec 5) is weak. Study basically finds that the effects, if there are any, in most cases are too small to be resolved from the noise. If the effect sizes are too small to be physiologically meaningful (as I gather) please state so directly. If the authors see the possibility of physiologically significant effects it is OK to discuss that but justify your reasoning. Please don't just write that EMF "could possibly" benefit "some performance outcomes".
Response: We appreciate the valuable input and have revised both the conclusions and the entire manuscript to align with the study’s findings more accurately.
Abstract (page 1, lines 27-31):
“In conclusion, ELF-EMF showed no significant overall effects on performance and inflammation after intense cycling among athletes. Post-hoc analysis revealed modest benefits of ELF-MLF, suggesting a context-dependent impact. Further research with larger samples and multiple sessions is needed to confirm ELF-EMF's recovery potential.”
Discussion (page 13, lines 466-473):
“In summary, our study did not find statistically significant overall effects of ELF-EMF on performance and blood inflammation following single high-intensity cycling among highly trained athletes. However, post-hoc analyses revealed modest, yet noteworthy benefits, indicating that the impact of ELF-EMF might be context-specific. These findings suggest potential positive effects in specific scenarios. Future research endeavors should aim to enhance the robustness of the findings regarding the advantageous effects of ELF-EMF as a recovery strategy by incorporating a more extensive sample size and conducting multiple exercise sessions.”
Minor:
line 239 says "nine trained athletes" but the table indicates n=10. Error?
Response: We apologize for the inconsistency of the sample size. We now clarify that ten participants were recruited but only nine participants were completed the all the five visits, the error has been rectified in Table 2 and in the appropriate lines in the abstract, method and results.
Abstract: “Nine male athletes (26.7±6.0y; 69.6±7.7kg, VO2peak 57.3±6.8ml/kg/min) completed five visits in a double-blinded cross-over design, performing two consecutive testing days, following a ventilatory thresholds assessment.” (Page 1 lines 18-20).
Method: “Ten healthy competitive male cyclists and triathletes were recruited to participate in the study.” (page 2 lines 83-84).
Results: “Ten trained male athletes were recruited to participate in the study. However, nine (26.7±6.0y; 69.6±7.7kg, VO2peak 57.3±6.8ml/kg/min) completed the five study visits and one participant completed only the baseline measurements (Table 2).” (page 7 lines 274-276).
please clarify in the figures, does "active" mean that the mat was generating magnetic fields? (I think so but am not sure). Add that to a figure caption.
Response: According the reviewer comment, we corrected Figure 1 in accordance with the terminology we defined for the two conditions in the manuscript: “In a randomized and double-blinded manner, two different conditions were applied for each participant: active condition (A-condition) or non-active condition (NA-condition). The active condition was when the ELF-EMF mattress was emitting electromagnetic fields.” (page 3, lines 108-111).
Should not results with an inactive mat at any particular time be the proper control? If that is the case, then the study is almost entirely negative.
Response: We did compare between the active and the non-active condition, but also the time effect separately to each condition. While our study did not find significant overall effects, modest yet noteworthy benefits were found from some of the data analysis. As described, the conclusions were modified accordingly.
Please don't use the term "significant" without a proper modifier to remove ambiguity. I think you mean "statistically significant" but readers are likely to understand that as "physiologically significant" - not proven here.
Response: We thank you for this comment. All the “significant” statements within the manuscript are now anchored to significant numbers within the results section. We have also added the word “statistically” to the first sentence of the conclusion.
in the introduction, the list of other studies reporting "effects" of ELF-EMF, without stating exposure strength or magnitude, is not informative. Better to replace this with a table listing the studies, endpoint, exposure characteristics, and outcome.
Response: Thank you for your comment. Our study includes many parameters and statistics, and in order not to burden the readers, we have added to the introduction the exposure strength or magnitude of previous studies’ ELF-EMF findings where pertinent and without adding another table. (page 2, line 57).
Reviewer 2 Report
“The effect of extremely low-frequency electromagnetic fields on inflammation and performance-related indices in trained athletes: a double-blinded crossover study”
Overall strengths of the article:
This manuscript explores the effects of Extremely Low-Frequency Electromagnetic Field (ELF-EMF) stimulation on post-high-intensity exercise recovery in trained athletes. Physiological indices, including physical performance, biochemical markers, inflammatory responses, pain perception, and fatigue rates, were assessed. The results revealed that ELF-EMF stimulation led to noticeable improvements in recovery rates for specific parameters, particularly in younger athletes with lower body fat. However, the overall effect size of ELF-EMF on most physiological indices remained relatively small. Despite this, the study suggests that ELF-EMF could play a modest role in improving muscle strength during daily high-intensity exercise by facilitating the recovery process. This research provides valuable insights into the application of non-invasive methods, such as ELF-EMF stimulation, to enhance recovery in sports. The observed improvements in performance-related indices and inflammatory responses may hold significance for athletes seeking to optimize their training outcomes.
An interesting study, however, it suffers from several major limitations that should be addressed before publication. Details are in the specific comments section.
Specific comments on weaknesses:
Major concerns:
1. No clear hypothesis and aims are presented in the abstract. It wasn’t clear what they wanted to test in this article. It was not clearly presented until the discussion section where they indicated what they wanted to do in this study. Need extensive rewriting.
2. Poore study design and experimental protocol: They have justified why specific frequencies were used but no clear explanation as to why specific tests and measurements are needed. What are the regions for selecting specific time points of measurement? Any inclusions and exclusions criteria?
3. The whole manuscript is poorly written with confusing sentences, for example: “This study has several limitations that should be acknowledged. Another limitation is the relatively short follow-up period of only 24 hours following high-intensity exercise”. What are those several limitations? Don’t you think a reader wants to know?
4. Extensive English correction throughout the manuscript is required.
Minor points:
1. FIGURE 5: Legend; “* Significant (p < 0.05) difference between conditions at. * Significant (p < 0.05) difference between conditions,” I’m not sure what is the difference between these two *.
2. Poor reference format and style, for example following three references are presented differently and should be presented in a better format.
1. A, Barrero, F S, G C, G K, D M, F C, et al. Daily fatigue-recovery balance monitoring with heart rate variability in well-trained female cyclists on the Tour de France circuit. PloS One [Internet]. 2019 Mar 7 [cited 2023 May 16];14(3). Available from: https://pubmed.ncbi.nlm.nih.gov/30845249/
2. Markus I, Constantini K, Hoffman JR, Bartolomei S, Gepner Y. Exercise-induced muscle damage: mechanism, assessment and nutritional factors to accelerate recovery. European Journal of Applied Physiology. Springer Science and Business Media Deutschland GmbH; 2021.
3. ROSS EZ, GREGSON W, WILLIAMS K, ROBERTSON C, GEORGE K. Muscle Contractile Function and Neural Control after Repetitive Endurance Cycling. Med Sci Sports Exerc. 2010;42(1).
Poorly written, Extensive English correction throughout the manuscript is required.
Author Response
We would like to thank you for handling the resubmission of our manuscript, entitled "The effect of extremely low-frequency electromagnetic fields on inflammation and performance-related indices in trained athletes: a double-blinded crossover study". We are grateful to the reviewers for their enlightening comments and important suggestions. In the revised version of the manuscript, we have addressed the specific points raised in their reviews, as described in more detail below in bold text and in the manuscript in red. We are confident that the alterations made in response to the reviewers' input will serve to elevate the overall quality and impact of our study.
Reviewer #2
An interesting study, however, it suffers from several major limitations that should be addressed before publication. Details are in the specific comments section.
Response: We thank the reviewer for the comments and the clear instructions how to revise and improve the manuscript.
Specific comments on weaknesses:
Major concerns:
- No clear hypothesis and aims are presented in the abstract. It wasn’t clear what they wanted to test in this article. It was not clearly presented until the discussion section where they indicated what they wanted to do in this study. Need extensive rewriting.
Response to point number 1: Thank you for this important comment. We have added aims and hypothesis to the manuscript in two sections:
Abstract (page 1 line 17-18)
“We aimed to shed light on ELF-EMF’s effect on recovery rate following high-intensity exercise.”
Introduction (page 2 lines 75-80)
“Considering the importance to athletes of having an accelerated recovery phase following multi-day competitions and the evidence of ELF-EMF’s therapeutic benefits, we aimed to examine the effect of ELF-EMF on recreationally trained cyclists’ recovery rates following acute, high intensity exercise. Our hypothesis was that ELF-EMF would attenuate the secretion of inflammatory and biochemical markers, blunt the response of performance-related indices, and reduce subjective soreness and fatigue.”
In addition, the rationale and operating hypotheses of the electromagnetic mattress was based on a vast literature review that cited in the methods materials which including more than twelve reference. (page 6 lines 230-235):
“According to Liboff and Zhadin the proposed explanation of the biological mechanism of ELF-EMF is that the coupling of a variable EMF (in the range applied in this study) with the geomagnetic field (GMF) creates the condition for the ion cyclotron resonance-like (ICR-like) effect [22,23]. Indeed, according to in vivo and in vitro studies, ELF-EMF frequencies have been demonstrated to induce various biochemical and physiological responses”.
- Poor study design and experimental protocol: They have justified why specific frequencies were used but no clear explanation as to why specific tests and measurements are needed. What are the regions for selecting specific time points of measurement? Any inclusions and exclusions criteria?
Response to point number 2: Thank you for the important comment. This study aimed to examine applicable physiological and subjective capabilities including the athletes’ rate of recovery while using an ELF-EMF mattress. For this, we added explanations and justifications for the measurements we used in this study: (page 3, lines 111-116, page 4, lines 133-134 & 161-162, 173-175, page 5 lines 187-188).
Inclusions and exclusions criteria were also added: “Ten healthy competitive male cyclists and triathletes were recruited to participate in the study. All participants were recruited based on their personal ability to perform 1-hour of cycling at an average power of at least 250w. Exclusion criteria included smokers and those taking prescribed medications or having a self-reported history of chronic pulmonary, cardiac, metabolic, or orthopedic conditions.” (page 2, lines 83-87).
The rationale behind the placement of the chosen timeframes was to facilitate an examination of the protocol's impact both with and without the treatment subsequent to the intense exercise regimen. The primary objective was to emulate practical scenarios of magnetic resonance mattress usage (the active intervention), such as after a race day or a multi-day competition. Our aim was to assess the effectiveness of the magnetic resonance intervention, whether in terms of immediate post-race short exposure or extended exposure periods.
- The whole manuscript is poorly written with confusing sentences, for example: “This study has several limitations that should be acknowledged. Another limitation is the relatively short follow-up period of only 24 hours following high-intensity exercise”. What are those several limitations? Don’t you think a reader wants to know?
Response to point number 3: We thank the author for this comment. Accordingly, the entire manuscript has been edited by a scientific editor, including the limitation section:
“The study has several limitations. First, there was a relatively brief 24-hour follow-up period subsequent to the high-intensity exercise protocol. This was because we wanted to emulate practical scenarios of ELF-EMF mattress usage (the active intervention), such as after a race day or a multi-day competition and assess the device’s effectiveness in terms of immediate post-race short exposure. However, this restricted duration might not adequately capture the entirety of the potential physiological and biochemical shifts that could manifest over a more extended timeframe. Therefore, a lengthier follow-up span of several days may provide more information. This links to the second study limitation. Namely, the study design included a constraint arising from the athletes' competitive commitments, thereby limiting the magnetic resonance exposure following the high-intensity protocol. Although this design constraint was driven by the athletes' competitive engagements, it might have limited the effect size. In future investigations, it might prove beneficial to extend both the magnetic intervention period and the subsequent follow-up window. Lastly, while for most study outcomes the all data was available and included in the final analysis, for IL-10, due to the performance of the ELISA kit, the sample size was smaller.” (page 13, lines 449 – 464)
- Extensive English correction throughout the manuscript is required.
Response to point number 4: A professional editor extensively proofreads the manuscript.
Minor points:
- FIGURE 5: Legend; “* Significant (p < 0.05) difference between conditions at. * Significant (p < 0.05) difference between conditions,”I’m not sure what is the difference between these two *.
Response to point number 1 (minor): According to this response we modified all figure in the next:
- Figure 2: * Significant (p < 0.05) time effect stratified by condition. # significant condition effect for the change at 60P in the post-hoc analysis. (page 8, lines 300-302).
- Figure 3: * Significant (p < 0.05) time effect stratified by condition. (page 9, lines 319-320).
- Figure 4: * Significant (p < 0.05) condition effect, stratified by cluster. (page 10, lines 352-353).
Figure 5: * Significant (p < 0.05) time effect stratified by condition, # Significant (p < 0.05) cluster*condition stratified by time. (page 11, lines 373-374).2. Poor reference format and style, for example following three references are presented differently and should be presented in a better format.
- A, Barrero, F S, G C, G K, D M, F C, et al. Daily fatigue-recovery balance monitoring with heart rate variability in well-trained female cyclists on the Tour de France circuit. PloS One [Internet]. 2019 Mar 7 [cited 2023 May 16];14(3). Available from: https://pubmed.ncbi.nlm.nih.gov/30845249/
- Markus I, Constantini K, Hoffman JR, Bartolomei S, Gepner Y. Exercise-induced muscle damage: mechanism, assessment and nutritional factors to accelerate recovery. European Journal of Applied Physiology. Springer Science and Business Media Deutschland GmbH; 2021.
- ROSS EZ, GREGSON W, WILLIAMS K, ROBERTSON C, GEORGE K. Muscle Contractile Function and Neural Control after Repetitive Endurance Cycling. Med Sci Sports Exerc. 2010;42(1).
Response to point number 2 (minor): Thank you. Corrected.
Round 2
Reviewer 1 Report
Authors have satisfactory responded to my comments. Recommend acceptance..
Reviewer 2 Report
The authors appear to have taken the reviews seriously and made the requested changes indicated by the reviewer. The overall article appears to be of high significance. There is still a large number of minor English errors that should be corrected, but that should be relatively simple.
Minor English errors that should be corrected.